# Fractal Calibration for Long-tailed Object Detection

## Abstract

Real-world datasets follow an imbalanced distribution, which poses significant challenges in rare-category object detection. Recent studies tackle this problem by developing re-weighting and re-sampling methods, that utilise the class frequencies of the dataset. However, these techniques focus solely on the frequency statistics and ignore the distribution of the classes in image space, missing important information. In contrast to them, we propose **Fra**ctal **CAL**ibration (FRACAL): a novel post-calibration method for long-tailed object detection. FRACAL devises a logit adjustment method that utilises the fractal dimension to estimate how uniformly classes are distributed in image space. During inference, it uses the fractal dimension to inversely downweight the probabilities of uniformly spaced class predictions achieving balance in two axes: between frequent and rare categories, and between uniformly spaced and sparsely spaced classes. FRACAL is a post-processing method and it does not require any training, also it can be combined with many off-the-shelf models such as one-stage sigmoid detectors and two-stage instance segmentation models. FRACAL boosts the rare class performance by up to $8.6\%$ and surpasses all previous methods on LVIS dataset, while showing good generalisation to other datasets such as COCO, V3Det and Open-Images. We provide the code in the Appendix.

## 1 Introduction

In recent years, there have been astonishing developments in the field of object detection Carion et al. (2020); Chen et al. (2022); Lyu et al. (2022). Most of these works utilise vast, balanced, curated datasets such as ImageNet1k Deng et al. (2009), or MS-COCO Lin et al. (2014) to learn efficient image representations. However, in the real world, data are rarely balanced, in fact, they follow a long-tailed distribution Liu et al. (2019). When models are trained with long-tailed data, they perform well for the frequent classes of the distribution but they perform inadequately for the rare classes Wang et al. (2020); Ren et al. (2020); Li et al. (2020). This problem poses significant challenges to the safe deployment of detection and instance segmentation models in real-world safe-critical applications such as autonomous vehicles, medical applications, and industrial applications, scenarios where rare class detection is paramount.

Many approaches address the long-tailed detection problem by employing adaptive re-weighting or data resampling techniques to handle imbalanced distributions Wang et al. (2021a;b); Zang et al. (2021). However all these methods require training. In contrast, in long-tailed image classification, alternative methods focus on mitigating class imbalance during inference through a post-calibrated softmax adjustment (PCSA) Alexandridis et al. (2023); Ren et al. (2020); Hong et al. (2021). PCSA boasts strong performance, good compatibility with many methods like data augmentation, masked image modeling, contrastive learning, and does not necessitate specialized loss function optimization, making it more user friendly Xu et al. (2023); Cui et al. (2021); Zhu et al. (2022). However, current PCSA methods utilise solely the train set's class frequency $p_s(y)$ as shown in Figure 1-left and they overlook the significance of the classes' dependence on the location distribution $p_s(y, u)$. This is a significant limitation of previous PCSA methods because the location information is a critical indicator considering the correlation between classes $y$ and their respective locations $u$.

Motivated by the class-location dependence Kayhan & Gemert (2020), in this work, we investigate a novel way to incorporate location information into post-calibration for imbalanced object detection

Figure 1: Previous PCSA used the class prior $p_s(y)$ to align the learned source distribution $p_s(y, u|x)$ with the balanced target distribution $p_t(y|x)$, without considering the space information $u$, highlighted in blue. FRACAL embeds space information $p_s(y, u)$ into class calibration, via the fractal dimension and aligns the learned $p_s(y, u|x)$ with $p_t(y, u|x)$ better than previous works.

to boost the performance of rare classes by fully exploiting dataset statistics. We empirically show that naively injecting location statistics results in inferior performance because the location information is sparse for the rare classes. To overcome this, we propose FRACAL (FRActal CALibration), a novel post-calibration method based on the fractal dimension, as shown in Figure 1-right. Our method aggregates the location distribution of all objects in the training set, using the box-counting method Schroeder (2009). This resolves the sparsity problem and significantly enhances the performance of both frequent and rare classes as shown in our experiments.

Our method comes with several advantages. First, it performs an effective class calibration, suitable for the object detection task, using the dataset's class frequencies. Secondly, it captures the class-location dependency Kayhan & Gemert (2020), using the fractal dimension, and it fuses this information into class calibration. This results in a better and unique space-aware logit-adjustment technique that complements the frequency-dependent class calibration method and achieves higher overall performance compared to previous PCSA techniques.

FRACAL can be easily combined with both one-stage and two stage detectors, Softmax and Sigmoid-based models, various instance segmentation architectures, various backbones, sampling strategies, and largely increase the performance during the inference step. FRACAL significantly advances the performance on the challenging LVISv1 benchmark Gupta et al. (2019) with no training, or additional inference cost by $8.6\%$ rare mask average precision ($AP_r^m$).

Our **contributions** are as follows:

- For the first time, we show the importance of the class-location dependence in post-calibration for long-tailed object detection.
- We capture the location-class dependence via a space-aware long-tailed object detection calibration method based on the fractal dimension.
- Our method performs remarkably on various detectors and backbones, on both heavily imbalanced datasets such as LVIS and less imbalanced datasets such as COCO, V3DET and OpenImages, outperforming the state-of-the-art by up to $8.6\%$.

## 2 RELATED WORK

**General Object Detection.** General object detection Redmon & Farhadi (2017); Ren et al. (2015); Lin et al. (2017b); Liu et al. (2016); Carion et al. (2020); Zhu et al. (2021); Sun et al. (2021); Chen et al. (2022); Li et al. (2022e) and instance segmentation He et al. (2017); Huang et al. (2019); Cai & Vasconcelos (2019); Chen et al. (2019a); Wang et al. (2019); Bolya et al. (2019); Li et al. (2022e) have witnessed tremendous advancements. Recently, transformer-based detectors were proposed which use self-attention to directly learn object proposals Carion et al. (2020); Zhu et al. (2021), or diffusion-based methods which use a de-noising process to learn bounding boxes Chen et al. (2022) and segmentation masks Gu et al. (2022b). However, all of these methods struggle to learn the rare classes when trained with long-tailed data Gupta et al. (2019); Oksuz et al. (2020) due to the insufficient rare samples. To this end, FRACAL enhances the rare class performance using a space-aware logit adjustment that can be easily applied during inference.

**Long-tailed image classification.** In the past years, the long-tailed image recognition problem has received great attention, as demonstrated by many recent surveys Oksuz et al. (2020); Zhang et al.

Table 1: Post-calibration techniques in long-tailed tasks. $\tau$ and $\gamma$ are hyper-parameters, $bg$ is the background class, $\mu_y$ and $\varsigma_y$ are estimated class mean and standard deviation respectively. Compared to past works, FRACAL considers both frequency and space statistics as shown in Section 3.

| Method | Dependency | Adjustment |
|---|---|---|
| Log. Adj. Menon et al. (2021) | Frequency | $z'_y = z_y - \tau \log(p_s(y))$ |
| IIF Alexandridis et al. (2023) | Frequency | $z'_y = -z_y \cdot \log(p_s(y))$ |
| PC-Softmax Hong et al. (2021) | Frequency | $z'_y = z_y - \log(p_s(y)) + \log(p_t(y))$ |
| Norcal Pan et al. (2021) | Frequency | $p'_y = \frac{p_y/n_y^\gamma}{p_{bg} + \sum p_y/n_y^\gamma}, y \notin bg$ |
| LogN Zhao et al. (2022a) | Frequency | $z'_y = \frac{z_y - (\mu_y - \min_y(\mu_y))}{\varsigma_y}, y \notin bg$ |
| FRACAL (ours) | Space & Frequency | $z'_y = \mathrm{S}(\mathrm{C}(z_y))/\sum_{j=1}^{C+1} \mathrm{S}(\mathrm{C}(z_y))$ |

(2023b); Yang et al. (2022a) and newly created benchmarks Yang et al. (2022b); Tang et al. (2022); Gu et al. (2022a). In long-tailed classification, the works could be split into two groups, representation learning and classifier learning. Representation learning techniques aim to efficiently learn rare class features using oversampling Park et al. (2022); Hong et al. (2022); Zang et al. (2021), contrastive learning Li et al. (2022d); Zhu et al. (2022); Cui et al. (2023), using ensemble or fusion models Wang et al. (2021c); Li et al. (2022c;b); Cui et al. (2022); Aimar et al. (2023), knowledge distillation Li et al. (2022c); He et al. (2021); Li et al. (2021a), knowledge transfer Liu et al. (2019); Parisot et al. (2022); Zhu & Yang (2020), sharpness aware minimisation Zhou et al. (2023a;b); Ma et al. (2023) and neural collapse Li et al. (2023); Zhong et al. (2023); Liu et al. (2023). Classifier learning techniques aim to adjust the classifier in favour of the rare classes via decoupled training Kang et al. (2020); Zhang et al. (2021b); Hsu et al. (2023), margin adjustment Menon et al. (2021); Ren et al. (2020); Hong et al. (2021); Cao et al. (2019); Hyun Cho & Krähenbühl (2022); Zhao et al. (2022b); Alexandridis et al. (2023); Ye et al. (2020) and cost-sensitive learning Cui et al. (2019); Khan et al. (2017); Wang et al. (2017). Among these works, the Post-Calibrated Softmax Adjustment (PCSA) method Menon et al. (2021); Hong et al. (2021); Ma et al. (2024) distinguishes itself through both its strong performance and the absence of any training requirements. However, most of the classifier and representation learning techniques are hard to adopt in long-tailed object detection. This difficulty arises from the larger imbalance inherent in this task, amplified by the presence of the background class Mullapudi et al. (2021); Yang et al. (2022a). Moreover, the optimisation of models for this task becomes more complex due to multiple sources of imbalance such as batch imbalance, class imbalance and task imbalance as outlined in this survey Oksuz et al. (2020). For this reason, we develop FRACAL, which is a post-calibration method tailored to the long-tailed object detection task. Different from post-calibration classification methods Menon et al. (2021); Hong et al. (2021), FRACAL enhances the detection performance by leveraging class-dependent space information derived from the fractal dimension. Through space-aware logit-adjustment, FRACAL mitigates biases in both the detection's location and classification axes.

**Long-tailed object detection.** The most prevalent technique is adaptive rare class re-weighting, which could be applied using either the statistics of the mini-batch Hsieh et al. (2021); Tan et al. (2020); Wang et al. (2021b) or the statistics of the gradient Tan et al. (2021); Li et al. (2022a). Other works use adaptive classification margins based on the classifier's weight norms Wang et al. (2022); Li (2022), classification score Feng et al. (2021); He et al. (2022); Wang et al. (2021a), activation functions Alexandridis et al. (2022; 2024), group hierarchies Li et al. (2020); Wu et al. (2020) and ranking loss Zhang et al. (2023a). Many works use data resampling techniques Zang et al. (2021); Gupta et al. (2019); Kang et al. (2020); Feng et al. (2021); Wu et al. (2020) or external rare class augmentation Zhang et al. (2022; 2021a). All these works optimise the model on the long-tailed distribution and require the construction of a complicated and cumbersome training pipeline. In contrast, our method operates during the model's inference stage thus it is easier to use and less evasive to the user's codebase.

Norcal Pan et al. (2021) was the first method to apply a post-calibration technique in imbalanced object detection, achieving promising results without training the detector. They proposed to calibrate only the foreground logits using the train-set's label statistics and applied a re-normalisation step. LogN Zhao et al. (2022a) proposed to use the model's own predictions to estimate the class statistics and applied standardisation in the classification layer. However LogN, requires forward-passing the whole training set through the model to estimate the weights, thus it is slower than FRACAL, which

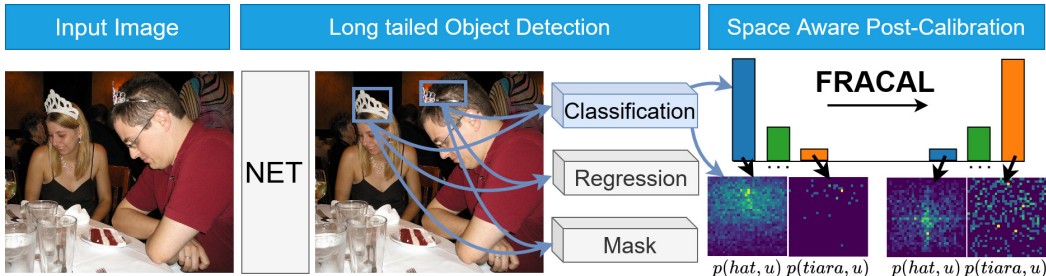

Figure 2: During imbalanced object detection, the model makes more frequent class predictions like *hat* and less rare class predictions like *tiara* both of which have strong upper location bias. FRACAL utilises fractal dimension and debiases the logits both in the frequency and space axes, making fewer *hat* predictions and more *tiara* predictions that are evenly spread, achieving space and frequency balance and increasing performance.

is not model-dependent. Also, both methods do not utilise the spatial statistics of the classes which are valuable indicators since the classes and their location are correlated Kayhan & Gemert (2020). To this end, FRACAL balances the detectors using both class and space information, largely surpassing the performance of the previous methods. FRACAL can be easily combined with two-stage softmax-based models like MaskRCNN He et al. (2017), or one-stage sigmoid detectors such as GFLv2 Li et al. (2021b) achieving great results without training or additional inference cost.

**Relation to previous works.** In Table 1, we contrast our work to previous post-calibration methods used in classification and object detection. As the Table suggests, all prior methods are frequency-dependent and none of them considers the space statistics.

## 3 METHODOLOGY

In Subsection 3.1, we pose the problem of calibration for classification; in Subsection 3.2, we extend it to the problem of object detection and we analyse the location-class dependence. We then, in Subsection 3.3, capture class-dependent space information via the fractal dimension and in Subsection 3.4, we combine it with the class-calibration method and extend it for binary object detectors. We show our approach in Fig. 2.

### 3.1 BACKGROUND: CLASSIFICATION CALIBRATION

Let $f_y(x; \theta) = z$ be a classifier parameterised by $\theta$, $x$ the input image, $y$ the class, $z$ the logit, $\bar{y}$ is the model's prediction and $p_s(y)$ and $p_t(y)$ the class priors on the train and test distributions respectively. The post-calibration equation is:

$$\bar{y} = \arg\max_y (f_y(x; \theta) + \log(p_t(y)) - \log(p_s(y))). \tag{1}$$

This has been numerously analysed in previous literature Menon et al. (2021); Alexandridis et al. (2023); Hong et al. (2021); Ren et al. (2020); Lipton et al. (2018) and we derive it in Appendix. In short, this shows that to get better performance, one can align the model's predictions with the test distribution, by subtracting $\log(p_s(y))$ and adding $\log(p_t(y))$ in the logit space. We now extend it to object detection.

### 3.2 CLASSIFICATION CALIBRATION FOR OBJECT DETECTION

In classification, $p(y)$ can be easily defined using the dataset's statistics, by using instance frequency $n_y$, i.e. $p(y) = \frac{n_y}{\sum_j^C n_j}$. In object detection, this is not the case because $p(y)$ is affected by the location and the object class. Accordingly, we define the class priors as:

$$p(y, o, u) = p(y|o, u) \cdot p(o, u) = p(y, u) \cdot p(o, u), \tag{2}$$

where $o$ denotes the generic object occurrence and $u$ is the location inside the image. By substituting Eq. 2 inside Eq. 1, we get $\bar{y}$ as:

$$\bar{y} = \arg\max_y (f_y(x; \theta) + \log(p_t(y, u) \cdot p_t(o, u)) - \log(p_s(y, u) \cdot p_s(o, u))). \tag{3}$$

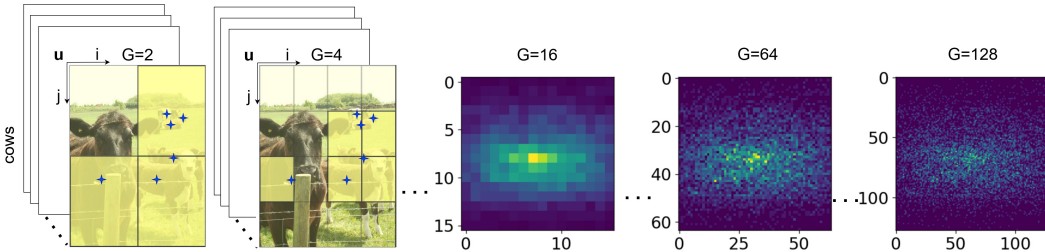

Figure 3: Different grid sizes affect the object distribution estimation. When the grid is coarse, e.g., $1 \times 1$ or $2 \times 2$, there is no or little location information. When it is finer, e.g., $128 \times 128$, the probability is sparse, giving noisy estimates for the rare classes.

The term $p(o, u)$ in Eq 3 cannot be calculated apriori as it depends on the model's training (e.g., the IoU sampling algorithm, how the object class is encoded etc[1]). Despite this, $p_s(o, u) \approx p_t(o, u)$, as we show in the Appendix, which means that the object distributions of the train and the test set remain the same and only the foreground class distribution changes. As a result:

$$\bar{y} = \arg\max_y (f_y(x; \theta) + \log(p_t(y, u)) - \log(p_s(y, u))) \tag{4}$$

Next, we show how the location parameter $u$ affects Eq. 4.

### 3.2.1 LOCATION-CLASS INDEPENDENCE.

First, we consider the scenario where the location $u$ does not give any information. In this scenario, $u$ and $y$ are independent variables, thus $p(y, u) = p(y) \cdot p(u)$ and we rewrite Eq. 4 as:

$$\begin{aligned}
\bar{y} &= \arg\max_y (f_y(x; \theta) + \log(p_t(y) \cdot p_t(u)) - \log(p_s(y) \cdot p_s(u))) \\
&= \arg\max_y (f_y(x; \theta) + \log(p_t(y)) - \log(p_s(y))),
\end{aligned} \tag{5}$$

where $p(u)$ is the probability of a random location in the image space and it has been simplified because it is the same in both source and target distributions, i.e., $p_s(u) = p_t(u)$.

In theory $p_t(y)$ is unknown, thus Eq.5 cannot be applied. Despite that, we found that setting $p_t(y) = \frac{1}{C}$ works well, because it forces the model to do balanced detections on the test set. In practice, this maximises average precision because this metric independently evaluates all classes and it rewards balanced detectors Everingham et al. (2010). Accordingly, the Classification (C) calibration of the logit $z_y$ is:

$$C(z_y) = \begin{cases} z_y - \log_\beta(\frac{n_y}{\sum_i^C n_i}) + \log_\beta(\frac{1}{C}), & y \in \{1, ..., C\} \\ z_y, & y = \text{bg}, \end{cases} \tag{6}$$

where $\beta$ is the base of the logarithm that we optimise through hyperparameter search. The background logit remains unaffected because of the assumption that the object distribution is the same in train and test set $p_s(o, u) \approx p_t(o, u)$, (this assumption is also found in previous works Pan et al. (2021); Zhao et al. (2022a)).

To this end, Eq. 6 can get good performance as shown in our ablation study but it is limited because the assumption that $p(y, u) = p(y) \cdot p(u)$ is not correct. In the real world, the object detection distribution has a strong center bias, as shown in Fig.3 and discussed in Oksuz et al. (2020). Furthermore, the location is correlated with the class Kayhan & Gemert (2020), therefore, $p(y, u) \neq p(y) \cdot p(u)$. As we show, the location provides valuable information for the long-tailed detection task and we enhance Eq. 6 by fusing location information.

### 3.2.2 LOCATION-CLASS DEPENDENCE.

One way to compute $p(y, u)$ is by counting the class occurrences $n_y(\mathbf{u})$ along locations that fall inside the cell $\mathbf{u} = [i, j]$ as shown in Fig. 3-left. To do so, we discretise the space of various image

---

[1]Typically object detectors use an extra background logit $bg$ to implicitly learn $p(o, u)$.

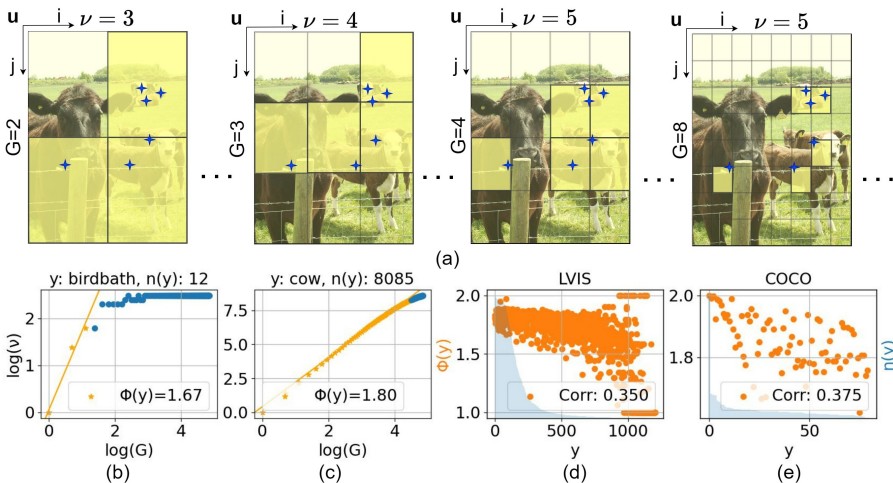

Figure 4: a) An example of the box counting method for the class *cow*. It iteratively counts the boxes containing its center $\nu$, as $G$ grows. b-c) The blue points are all $G - \nu$ pairs, out of them only the orange points are used to calculate the slope $\Phi$ based on the quadratic rule $G = \lfloor \sqrt{n_y} \rfloor$. d-e) Scatter-plot of $\Phi$ against instance frequency, there is a weak correlation i.e. 0.35 for LVISv1 and 0.375 for COCO using Pearson's correlation.

resolutions into a normalised square grid $U_{G \times G}$ of fixed size $G \in \mathbf{N}$ and count class occurrences inside every grid cell. Accordingly, the grid dependent calibration is defined as:

$$
\mathbf{C}_G(z_{y,\mathbf{u}}) = \begin{cases} z_{y,\mathbf{u}} - \log_\beta(p_s(y, \mathbf{u})) + \log_\beta(p_t(y, \mathbf{u})) \\ z_{y,\mathbf{u}}, \quad if \quad y = \text{bg}, \end{cases} \tag{7}
$$

where $z_{y,\mathbf{u}}$ is the predicted proposal whose center falls inside the discrete cell $\mathbf{u} = [i, j]$ and $p_t(y, \mathbf{u})$ is uniform, i.e., $p_t(y, \mathbf{u}) = \frac{1}{C} \cdot \frac{1}{G^2}$.

However, the choice of the grid size $G$ largely affects the estimation of $p(y, u)$, as shown in Fig. 3-right. For example, if we use smaller $G$, the generic object distribution becomes denser and little location information is encoded. If we use larger $G$, the distribution becomes sparse. This is problematic for the rare classes because they are already sparse and their location information is noisy. In Table 4-e, we show that this method shows limited performance.

## 3.3 CALIBRATION USING FRACTALS

To solve the sparsity problem introduced by the grid-size, we use fractal dimension $\Phi$ Panigrahy et al. (2019), which is a metric independent of the grid size $G$. To calculate $\Phi$, we use the box-counting method Schroeder (2009):

$$
\Phi(y) = \lim_{G \to \infty} \frac{\log \sum_{j=0}^{G-1} \sum_{i=0}^{G-1} \mathbb{1}(n_y(\mathbf{u}))}{\log(G)}, \tag{8}
$$

where $\mathbb{1}$ is the indicator function. For objects in 2D images, as in our case, $\Phi(y) \in [0, 2]$, where 0 is only one object, 1 shows that the objects lie across a line and 2 shows that they are located uniformly across the image space.

For brevity, we rewrite $\nu_y = \sum_{j=0}^{G-1} \sum_{i=0}^{G-1} \mathbb{1}(n_y(\mathbf{u}))$ and we give an example in Fig. 4-a. In practice, Eq. 8 cannot be computed because by increasing $G$, the computation becomes intractable. Instead, we approximate $\Phi$, by evaluating nominator-denominator pairs of Eq. 8 for various values of $G$ up to a threshold $t$ and then fit a line to those points. The slope of this line approximates $\Phi(y)$, because it considers all computed $G - \nu_y$ pairs.

**Dealing with rare classes.** To select the threshold $t$, we use the quadratic rule $G \leq t = \lfloor \sqrt{n_y} \rfloor$. The motivation for this rule is simple, for example, if an object is rare, e.g., it appears 4 times in the whole training set, then it can, at most, fill a grid of size $2 \times 2$. For objects with fewer occurrences *we cannot compute $\Phi$ and thus we assign $\Phi = 1$*. Using this rule, we define the maximum number of

pairs that are required for fitting the "fractality" line highlighted in orange in Fig. 4-b and Fig. 4-c. For example, the rare object *birdbath* appears 12 times in the training set, thus we use the first three orange points in Fig. 4-c that correspond to $G = \{1, 2, 3\}$, to fit the "fractality" line, resulting in a large $\Phi = 1.67$. This rule ensures that the fractal dimension computation does not underestimate the rare classes and it gives robust measurements that increase rare class performance as shown in our experiments. For the *cow* object that has larger frequency we use more $G - \nu$ orange pairs to fit the line as shown in Fig. 4-b, resulting in $\Phi = 1.80$.

**Relationship to frequency.** As shown in Fig. 4-d, the fractal dimension weakly correlates with frequency for the LVISv1 dataset, i.e., $0.35$ using Pearson correlation. However, there are many rare classes with large $\Phi \approx 2$, showing that our threshold selection technique is robust for small sample sets. Also, the correlation for the COCO dataset in Fig. 4-e is similar to LVIS because these datasets have the same images. This shows that the relationship of the estimated fractal dimension and the frequency is dependent on the image data itself and not the class imbalance and it highlights that our method is robust for different label distributions, it is not a purely frequency-based method, and it captures the class space statistics effectively.

**Usage.** After calculating $\Phi$ for all classes in the training set, we perform Space-dependent class calibration (S) during inference:

$$\text{S}(z_y) = \begin{cases} \frac{\sigma(z_y)}{\Phi(y)^\lambda}, & y \in \{1, ..., C\} \\ \sigma(z_y), & y = \text{bg}, \end{cases} \tag{9}$$

where $\sigma(z_y) \in (0, 1)$ is the model's prediction for class $y$, with $\sigma()$ the Softmax activation, and $\lambda \geq 0$ is a hyperparameter. Eq. 9 downweighs the classes that appear most uniformly and it upweighs the classes that appear less uniformly. In practice, this scheme enforces a centre bias for frequent classes and no bias for rare classes, as shown in Fig. 2-bottom-right. Intuitively, removing the bias from the rare classes is better than rectifying it because it produces balanced detectors and aligns better with the target distribution as shown in our ablation and our qualitative results.

### 3.4 LOCALISED CALIBRATION

By putting Eq. 6 and Eq. 9 together, we get the final FRActal CALibration (FRACAL) as:

$$\text{FRACAL}(z_y) = \frac{\text{S}(\text{C}(z_y))}{\sum_{j=1}^{C+1} \text{S}(\text{C}(z_j))}. \tag{10}$$

Our proposed method tackles the classification imbalance using additional space statistics. On the classification axis, we use the class priors $p_s(y)$ and perform logit adjustments. On the space axis, we use the fractal dimension $\Phi(y)$ to perform a space-aware calibration that accounts for the object's location distribution $p_s(y, u)$. In Eq. 10, we renormalise both foreground and background logits to preserve a probabilistic prediction after the space calibration in Eq. 9.

**Extending to binary classifiers.** In long-tailed object detection there are many works that use only binary classifiers Alexandridis et al. (2022); Tan et al. (2020; 2021); Li et al. (2022a); Wang et al. (2021b); Hyun Cho & Krähenbühl (2022); Hsieh et al. (2021). In this case, the logit $z_i$ performs two tasks simultaneously: It discriminates among the foreground classes and performs background-to-foreground classification. Thus, to correctly apply foreground calibration, we first need to decouple the foreground and background predictions. To do so, we filter out the background proposals using the model's predictions as follows:

$$\text{FRACAL}_b(z_i) = \eta(\text{C}(z_i) - \log_\beta(\frac{\Phi(y)^\lambda}{\sum_i^C \Phi(i)^\lambda}) + \log_\beta(\frac{1}{C})) \cdot \eta(z_i), \tag{11}$$

where $\eta(z_i)$ is the sigmoid activation function that acts as a filter for low-scoring proposals. Compared to Eq. 10, Eq. 11 performs class calibration and space calibration in logit space, lowering the false-positive detection rate.

## 4 RESULTS

### 4.1 EXPERIMENTAL SETUP

We use the Large Vocabulary Instance Segmentation (LVISv1) dataset Gupta et al. (2019) which consists of $100k$ images in the train set and $20k$ images in the validation set. This dataset has $1,203$

Table 2: Comparison against SOTA on LVISv1 dataset. All competing methods use MaskRCNN and our method reaches the best results in all metrics. † denotes our re-implementation with RFS.

| Method | Arch. | $AP^m$ | $AP_r^m$ | $AP_c^m$ | $AP_f^m$ | $AP^b$ |
|---|---|---|---|---|---|---|
| NorCal Pan et al. (2021) | | 25.2 | 19.3 | 24.2 | 29.0 | 26.1 |
| LogN Zhao et al. (2022a) | | 27.5 | 21.8 | 27.1 | 30.4 | 28.1 |
| GOL Alexandridis et al. (2022) | R50 | 27.7 | 21.4 | 27.7 | 30.4 | 27.5 |
| ECM Hyun Cho & Krähenbühl (2022) | | 27.4 | 19.7 | 27.0 | 31.1 | 27.9 |
| CRAT w/ LOCE Wang et al. (2024) | | 27.5 | 21.2 | 26.8 | 31.0 | 28.2 |
| **FRACAL (ours)** | | **28.6** | **23.0** | **28.0** | **31.5** | **28.4** |
| NorCal Pan et al. (2021) | | 27.3 | 20.8 | 26.5 | 31.0 | 28.1 |
| LogN Zhao et al. (2022a) | | 29.0 | 22.9 | 28.8 | 31.8 | **29.8** |
| GOL Alexandridis et al. (2022) | | 29.0 | 22.8 | 29.0 | 31.7 | 29.2 |
| ECM Hyun Cho & Krähenbühl (2022) | R101 | 28.7 | 21.9 | 27.9 | 32.3 | 29.4 |
| ROG Zhang et al. (2023a) | | 28.8 | 21.1 | 29.1 | 31.8 | 28.8 |
| CRAT w/ LOCE Wang et al. (2024) | | 28.8 | 22.0 | 28.6 | 32.0 | 29.7 |
| **FRACAL (ours)** | | **29.8** | **24.5** | **29.3** | **32.7** | **29.8** |

classes grouped according to their image frequency into *frequent* (those that contain $> 100$ images), *common* (those that contain $10 \sim 100$ images) and *rare* classes (those that contain $< 10$ images) in the training set. For evaluation, we use average mask precision $AP_m$, average box precision $AP_b$ and $AP_f^m$, $AP_c^m$ and $AP_r^m$ that correspond to $AP^m$ for *frequent*, *common* and *rare* classes. Unless mentioned, we use Mask R-CNN He et al. (2017) with FPN Lin et al. (2017a), ResNet50 He et al. (2016), repeat factor sampler (RFS) Gupta et al. (2019), Normalised Mask and cosine classifier as used in Wang et al. (2021a), CARAFE Wang et al. (2019) and we train the baseline model using the 2x schedule He et al. (2019), SGD, learning rate $0.02$ and weight decay $1e-4$. For Swin-T, we train the baseline model with the 1x schedule, RFS, AdamW Kingma & Ba (2014) and $0.001$ learning rate. During inference, we set the IoU threshold at $0.3$ and the mask threshold at $0.4$. FRACAL is applied before the non-maximum suppression step. We use the mmdetection framework Chen et al. (2019b) and train the models using V100 GPUs.

## 4.2 Main Results

**Comparison to SOTA.** In Table 2, we compare FRACAL to the state-of-the-art using ResNet50 and ResNet101. Using ResNet50, FRACAL significantly surpasses GOL Alexandridis et al. (2022) by $0.9$pp $AP^m$ and by $1.6$pp $AP_r^m$. On ResNet101 FRACAL achieves $29.8\%$ $AP^m$ and $24.5\%$ $AP_r^m$, outbesting GOL by $0.8$pp and $1.7$pp respectively.

FRACAL achieves excellent results not only for rare categories but also for frequent ones, due to the use of fractal dimension, which allows the model to upscale the predictions of frequent but non-uniformly located categories. It achieves $31.5\%$ $AP_f^m$ with ResNet50 and $32.7\%$ $AP_f^m$ with ResNet101 and surpasses the next best method, ECM Hyun Cho & Krähenbühl (2022) by $0.4$pp.

Compared to the previous post-calibration method, Norcal Pan et al. (2021), FRACAL increases performance by $3.4$pp $AP^m$, $3.7$pp $AP_r^m$, $3.8$pp $AP_c^m$, $2.5$pp $AP_f^m$ and $2.3$pp $AP^b$ using ResNet50. This is because FRACAL boosts both rare and frequent categories via classification and space calibration, respectively, while Norcal only boosts the rare categories and lacks space information.

We also compare our method in Transformer backbones. Using Swin-T, FRACAL considerably outperforms Seesaw Wang et al. (2021a) by $1.2$pp $AP^m$, $1.7$pp $AP_r^m$, $1.2$pp $AP_c^m$, $1.0$pp $AP_f^m$ and $0.8$pp $AP^b$ as shown in Table 3-a. Using Swin-S, FRACAL largely surpasses Seesaw in all metrics and particularly in $AP_r^m$ by $2.2$pp which is a significant $8.6\%$ relative improvement for the rare classes.

**Results on object detectors.** We evaluate FRACAL with common object detectors in Table 3-b using ResNet50. FRACAL boosts the overall and rare category performance of both one-stage detectors such as ATSS Zhang et al. (2020) or GFLv2 Li et al. (2021b) and two-stage detectors such as Cascade RCNN Cai & Vasconcelos (2019) and APA-MaskRCNN Alexandridis et al. (2024). Note that on sigmoid-detectors such as ATSS or GFLv2, FRACAL largely boosts the performance of rare and common categories but it slightly reduces the performance of frequent categories. Since

Table 3: In (a), we show that FRACAL can be used with Swin transformers Liu et al. (2021) and surpass the SOTA. In (b), we show that FRACAL can be used with both Sigmoid and Softmax based detectors and improve their precision.

| Method | $AP^m$ | $AP^m_r$ | $AP^m_c$ | $AP^m_f$ | $AP^b$ |
|---|---|---|---|---|---|
| RFS-(T) | 27.7 | 17.9 | 27.9 | 31.8 | 27.1 |
| Seesaw-(T) | 29.5 | 24.0 | 29.3 | 32.2 | 29.5 |
| GOL-(T) | 28.5 | 21.1 | 29.5 | 30.6 | 28.3 |
| **FRACAL-(T)** | **30.7** | **25.7** | **30.5** | **33.2** | **30.3** |
| RFS-(S) | 30.9 | 21.7 | 31.0 | 34.7 | 31.0 |
| Seesaw-(S) | 32.4 | 25.6 | 32.8 | 34.9 | 32.9 |
| GOL-(S) | 31.5 | 24.1 | 32.3 | 33.8 | 32.0 |
| **FRACAL-(S)** | **33.6** | **27.8** | **33.9** | **35.9** | **33.4** |

(a) Results using Swin (T/S) and MaskRCNN.

| Method | $AP^b$ | $AP^b_r$ | $AP^b_c$ | $AP^b_f$ |
|---|---|---|---|---|
| ATSS Zhang et al. (2020) | 25.3 | 15.8 | 23.4 | 31.6 |
| **w/ FRACAL (ours)** | **26.7** | **20.8** | **25.9** | 30.9 |
| GFLv2 Li et al. (2021b) | 26.6 | 14.7 | 25.1 | 33.5 |
| **w/ FRACAL (ours)** | **28.2** | **19.4** | **27.2** | 33.2 |
| GFLv2 (DCN) Li et al. (2021b) | 27.4 | 13.7 | 26.1 | 34.8 |
| **w/ FRACAL (ours)** | **28.9** | **18.7** | **27.9** | 34.5 |
| APA Alexandridis et al. (2024) | 26.9 | 14.3 | 26.2 | 33.2 |
| **w/ FRACAL (ours)** | **29.2** | **22.1** | **28.0** | **33.7** |
| C-RCNN Cai & Vasconcelos (2019) | 28.6 | 16.5 | 27.8 | 34.9 |
| **w/ FRACAL (ours)** | **31.5** | **24.3** | **31.0** | **35.3** |

(b) Comparisons using various detectors and ResNet50.

Table 4: Ablations using MaskRCNN-ResNet50. C and S denote the class and location calibration.

| C | S | $AP^m$ | $AP^m_r$ |
|---|---|---|---|
| | | 22.8 | 8.2 |
| | ✓ | 25.6 | 13.7 |
| ✓ | | 26.3 | 16.5 |
| ✓ | ✓ | **27.3** | **19.0** |

(a) Random sampler.

| λ | $AP^m$ | $AP^m_r$ | $AP^m_c$ | $AP^m_f$ | $AP^b$ |
|---|---|---|---|---|---|
| 0.0 | 28.0 | 22.4 | 27.3 | 31.2 | 27.4 |
| 1.0 | 28.5 | 23.0 | **28.0** | **31.6** | 28.3 |
| **2.0** | **28.6** | 23.0 | **28.0** | 31.5 | **28.4** |
| 3.0 | 28.5 | 23.2 | **28.0** | 31.5 | **28.4** |
| 4.0 | 28.5 | **23.4** | 27.9 | 31.3 | **28.4** |

(b) Ablation study of $\beta$, with RFS.

| $\beta$ | random | | RFS | |
|---|---|---|---|---|
| | $AP^m$ | $AP^m_r$ | $AP^m$ | $AP^m_r$ |
| 2 | 19.9 | 14.7 | 19.9 | 18.8 |
| $e$ | 25.1 | 16.6 | 25.8 | 21.1 |
| **10** | **26.3** | 16.5 | **28.0** | **22.4** |

(c) Ablation study of $\lambda$.

| C | S | $AP^m$ | $AP^m_r$ |
|---|---|---|---|
| | | 25.7 | 15.8 |
| | ✓ | 27.7 | 20.7 |
| ✓ | | 28.0 | 22.4 |
| ✓ | ✓ | **28.6** | **23.0** |

(d) Results using RFS.

| Method | $AP^m$ | $AP^m_r$ | $AP^m_c$ | $AP^m_f$ | $AP^b$ |
|---|---|---|---|---|---|
| G=1 | 28.0 | 22.4 | 27.3 | 31.2 | 27.4 |
| G=2 | 27.1 | 17.5 | 27.2 | 31.1 | 26.6 |
| G=4 | 25.0 | 10.5 | 25.4 | 31.1 | 24.9 |
| **ours** | **28.6** | **23.0** | **28.0** | **31.5** | **28.4** |

(e) Results with Grid-based method.

| Method | $AP^m$ | $AP^m_r$ | $AP^b$ |
|---|---|---|---|
| Invert FRACAL | 27.4 | 20.5 | 26.9 |
| Normal FRACAL | **28.6** | **23.0** | **28.4** |

(f) Invert FRACAL is inferior.

the sigmoid activation performs independent classification, the binary version of FRACAL struggles to properly calibrate the predicted unnormalised vector. This limitation was also found in previous works Pan et al. (2021) which also reported that binary logit adjustment produces performance trade-offs between frequent and rare categories. For softmax-based detectors, such as Cascade RCNN and APA, FRACAL boosts all categories.

## 4.3 Ablation Study and Analysis

**The effect of each module.** FRACAL consists of simple modules that we ablate in Table 4-a. First, MaskRCNN with CARAFE Wang et al. (2019), normalised mask predictor Wang et al. (2021a), cosine classifier Wang et al. (2021a) and random sampler achieves 22.8% $AP^m$ and 8.2% rare category $AP^m_r$. On top of this, the fractal dimension calibration (S) improves $AP^m$ and $AP^m_r$ by 2.8pp and 5.5pp respectively.

Using only the classification calibration, (C), $AP^m$ and $AP^m_r$ are enhanced by 3.5pp and 8.3pp respectively, because this technique majorly upweights the rare classes. When (S) is added, then it further increases $AP^m$ by 1.0pp and $AP^m_r$ by 2.5pp compared to only (C), reaching 27.3% $AP^m$ and 19.0% $AP^m_r$. This suggests that (S) is useful and the detector can benefit from space informa-tion. The same trend is observed with RFS in Table 4-d, however, both calibration methods have lower gains because RFS partly balances the classes via oversampling.

**Class calibration parameter search.** We further ablate the choice of the log base $\beta$ in Eq. 6, using the most common cases: 2 (bit), $e$ (nat), and 10 (hartley). As shown in Table 4-b, the base-10 is the best as it achieves 26.3% $AP^m$ and 16.5% $AP^m_r$ with the random sampler and 28.0% $AP^m$ and 22.4% $AP^m_r$ with RFS, thus we use it for all experiments on LVIS. We also observe that further increasing $\beta$ does not come with a performance improvement.

**Fractal dimension coefficient.** We ablate the choice of the $\lambda$ coefficient in the fractal dimension calibration Eq. 9. As shown in Table 4-c, the optimal performance is achieved with $\lambda = 2$ which increases the rare categories by 0.6pp, the common categories by 0.7pp, the frequent categories by 0.3pp, the overall mask performance by 0.6pp and the box performance by 1.0pp.

**Comparison to grid-dependent calibration.** We compare FRACAL against the grid-based method, Eq. 7, in Table 4-c. When $G = 1$ the method does not consider any location informa-tion because all predictions fall inside the same grid cell. This achieves the best performance and it

Table 5: Results on other detection datasets.

(a) Results on COCO with MaskRCNN.

| Method | $AP^m$ | $AP^b$ |
|---|---|---|
| ResNet50 He et al. (2016) | 35.4 | 39.4 |
| with FRACAL (ours) | **35.8** | **39.9** |
| SE-ResNet50 Hu et al. (2018) | 36.9 | 40.5 |
| with FRACAL (ours) | **37.4** | **41.1** |
| CB-ResNet50 Woo et al. (2018) | 37.3 | 40.9 |
| with FRACAL (ours) | **37.8** | **41.5** |
| Swin-T Liu et al. (2021) | 41.6 | 46.0 |
| with FRACAL (ours) | **41.9** | **46.4** |

(b) Results on V3Det Wang et al. (2023) with FasterRCNN and ResNet50.

| Method | $AP^b$ | $AP^b_{50}$ | $AP^b_{75}$ |
|---|---|---|---|
| APA Alexandridis et al. (2024) | 29.9 | 37.6 | 32.9 |
| with FRACAL (ours) | **30.3** | **37.7** | **33.2** |

(c) Results on OpenImages using ResNet50.

| Method | Detector | $AP^b_{50}$ |
|---|---|---|
| CAS Liu et al. (2020) | FRCNN | 65.0 |
| CAS with FRACAL (ours) | | **67.0** |
| CAS Liu et al. (2020) | CRCNN | 66.3 |
| ECM Hyun Cho & Krähenbühl (2022) | | 65.8 |
| CAS with FRACAL (ours) | | **67.5** |

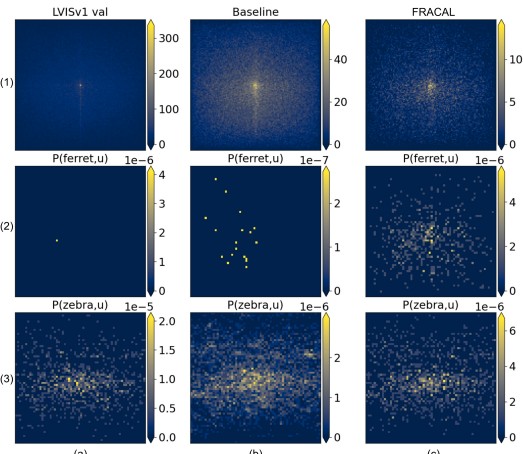

Figure 5: Detection results in LVIS-val. FRACAL detects more uniformly located rare classes in (2c) and less uniformly located frequent ones in (3c) than the baseline in (2b) and (3b).

is the same result with the $\lambda = 0$ of Table 4-c. When the grid size $G$ is enlarged, the performance of the rare classes drops significantly because the estimated prior distribution $p_s(y, \mathbf{u})$ becomes sparse (see Fig. 3). FRACAL does not suffer from this problem, because it re-weights all proposals based on fractal dimension.

**Inverting FRACAL.** We further examine the opposite weighting logic, which is to upweight the uniform located classes and downweight the non-uniform located classes. This technique further rectifies the location bias instead of removing it from the object detectors. As Table 4-f shows, the Invert FRACAL method is inferior to the normal one, because it produces an imbalanced detector.

**Generalisation to other datasets.** We test FRACAL on MS-COCO Lin et al. (2014), V3DET Wang et al. (2023) and OpenImages Kuznetsova et al. (2020) to understand its generalisation ability. The first two datasets are fairly balanced therefore, we do not expect our long-tailed designed detector to massively outperform the others. In Table 5 a-b, FRACAL increases the performance of all models, by an average of 0.5pp $AP^b$ and $AP^m$ on COCO and by 0.4pp $AP^b$ on V3DET. In Table 5-c, we show that FRACAL outperforms ECM using CascadeRCNN by 1.7pp and it further increases the performance of CAS by 2.0pp and 1.2pp using FasterRCNN and CascadeRCNN respectively.

**Qualitative Analysis.** In Fig. 5, we show: (a) the ground truth distribution, (b) the baseline and (c) FRACAL predicted distributions concerning general objects (1), the rare class *ferret* (2) and the frequent class *zebra* (3). FRACAL achieves better precision than the baseline because it predicts fewer generic objects in (1-c) than the baseline (1-b); it predicts more rare classes that are more uniformly located in (2-c) than the baseline in (2-b); and it predicts less frequent classes that have a stronger center-bias as shown in (3-c) than the baseline in (3-b). These results show that FRACAL aligns its predictions better with the ground-truth distribution than the baseline.

## 5 CONCLUSION

We propose FRACAL, a novel post-calibration method for long-tailed object detection. Our method performs a space-aware logit adjustment, that utilises the fractal dimension and incorporates space information during calibration. FRACAL majorly boosts the performance of the detectors and makes more rare class predictions that are evenly spread inside the image. We show that FRACAL can be easily combined with both one-stage Sigmoid detectors and two-stage Softmax segmentation models. Our method boosts the performance of detectors by up to $8.6\%$ without training or additional inference cost, surpassing the SOTA in the LVIS benchmark and showing good generalisation to COCO, V3Det and OpenImages.

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

## A  BACKGROUND: CLASSIFICATION CALIBRATION

We theoretically derive the classification calibration for image classification. Let $p_s(y|x)$ and $p_t(y|x)$ be the source and target conditional distributions. Using the Bayes theorem, we write the source and target conditional distributions as:

$$p_s(y|x) = \frac{p_s(x|y)p_s(y)}{p_s(x)} \ , p_t(y|x) = \frac{p_t(x|y)p_t(y)}{p_t(x)} \tag{12}$$

Dividing them, we write the target conditional distribution:

$$p_t(y|x) = \frac{1}{\kappa(x)} \frac{p_t(y)}{p_s(y)} p_s(y|x) \frac{p_t(x|y)}{p_s(x|y)} \tag{13}$$

where $\kappa(x) = \frac{p_t(x)}{p_s(x)}$. During training, we approximate $p_s(y|x)$ by model $f_y(x;\theta) = z$ and a scorer function $s(x) = e^x$ for multiple category classification. Thus, the learned source conditional distribution is $p_s(y|x) \propto e^{f_y(x;\theta)}$. Substituting it inside Eq. 13, we rewrite the target condition distribution as:

$$\begin{aligned} p_t(y|x) &\propto \frac{1}{\kappa(x)} \frac{p_t(y)}{p_s(y)} e^{f_y(x;\theta)} \frac{p_t(x|y)}{p_s(x|y)} \\ &= d(x,y) \cdot e^{f_y(x;\theta) + \log(p_t(y)) - \log(p_s(y)) - \log(\kappa(x))} \end{aligned} \tag{14}$$

where we assume that $d(x,y) = \frac{p_t(x|y)}{p_s(x|y)} = 1$. This is a reasonable assumption, in cases where both train and test generating functions come from the same dataset, as it is in our benchmarks. In inference, we calculate the prediction $\bar{y}$ by taking the maximum value of Eq. 14:

$$\begin{aligned} \bar{y} &= \arg\max_y e^{(f_y(x;\theta) + \log(p_t(y)) - \log(p_s(y)) - \log(\kappa(x)))} \\ &= \arg\max_y (f_y(x;\theta) + \log(p_t(y)) - \log(p_s(y))) \end{aligned} \tag{15}$$

where $\kappa(x)$ is simplified because it is a function of $x$ and it is invariant to $\arg\max_y$. Eq. 15 is the post-calibration method Menon et al. (2021); Hong et al. (2021). It can be used during inference to achieve balanced performance by injecting prior knowledge inside the model's predictions, via $p_t(y)$ and $p_s(y)$, in order to align the source with the target label distribution and compensate for the label shift problem.

## B  FRACTAL DIMENSION VARIANTS

We explore various ways for computing the fractal dimension using the box-counting method Schroeder (2009), the information dimension Rényi (1959) (Info), and a smooth variant (Smooth-Info). The information variant is defined as:

$$\text{Info-}\Phi(y) = \lim_{G \to \infty} \frac{\log \sum_{j=0}^{G-1} \sum_{i=0}^{G-1} \frac{\mathbb{1}(n_y(\mathbf{u}))}{G}}{\log(G)} \tag{16}$$

| Dimension | $AP^m$ | $AP^m_r$ | $AP^b$ |
|:---:|:---:|:---:|:---:|
| Info | **28.6** | 23.2 | 28.3 |
| SmoothInfo | **28.6** | **23.4** | 28.3 |
| **Box** | **28.6** | 23.0 | **28.4** |

Table 6: Fractal Dimension Variants using MaskRCNN with ResNet50 and RFS on LVISv1. All of the are robust and we have chosen the Box variant in the main paper.

It is the similar to the box-counting dimension, except for the box count which is normalised by dividing by the grid size $G$. This way, the information dimension is represented by the growth rate of the probability $p = \frac{\mathbb{1}(n_y(\mathbf{u}))}{G}$ as $G$ grows to infinity.

In practise, the quantity $\mathbb{1}(n_y(\mathbf{u}))$ can be frequently zero for many locations $\mathbf{u}$ especially for rare classes that have few samples and are sparsely located. For this reason, we also proposed a smooth information variant defined as:

$$\text{Smooth-}\Phi(y) = \lim_{G \to \infty} \frac{1 + \log \sum_{j=0}^{G-1} \sum_{i=0}^{G-1} \frac{1 + \mathbb{1}(n_y(\mathbf{u}))}{G}}{\log(G)} \tag{17}$$

This Equation is inspired by the smooth Inverse Document Frequency Robertson (2004) used in natural language processing and its purpose is to smooth out zero values in $\mathbb{1}(n_y(\mathbf{u}))$ calculation.

All variants are robust and SmoothInfo achieves slightly better $AP^m_r$ because its calculation is more tolerant to few samples compared to the box-counting method. However, SmoothInfo and Info achieve slightly worse $AP^b$, thus we use the box-counting method in the main paper.

## C  OBJECT DISTRIBUTIONS

We show that the object distribution $p_s(o, u)$ in the training set is similar to the object distribution $p_t(o, u)$ on the test set in the LVIS v1 dataset Gupta et al. (2019). As shown in Figure 6, the distributions are close therefore we can safely assume that $p_s(o, u) \approx p_t(o, u)$. This explains the reason why the background logit should remain intact during calibration because there does not exist label shift for the generic object class (also for the background class) between the train and test sets.

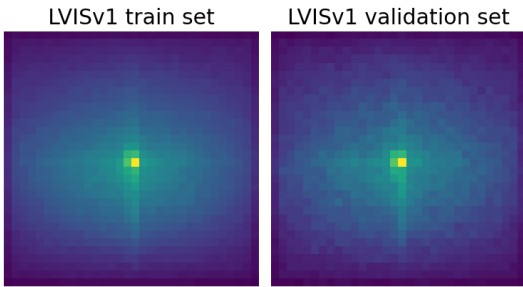

Figure 6: Comparison between the $p_s(o, u)$ (left) and $p_t(o, u)$ (right) in LVISv1 dataset. The distributions are similar, therefore we can safely assume that $p_s(o, u) \approx p_t(o, u)$.

