# OpenReview forum: "FRACTAL CALIBRATION FOR LONG-TAILED OBJECT DETECTION"
_ICLR.cc/2025/Conference — ICLR 2025 Conference Withdrawn Submission_

### Official Review · Reviewer_b1YX · 2024-10-27

**Soundness:** 2
**Presentation:** 3
**Contribution:** 2
**Rating:** 5
**Confidence:** 4

**Summary:**

In this paper, the authors introduce a calibration method for object detectors. Unlike prior work focusing on this problem, the authors propose considering the location distribution of the samples as well while adjusting the probabilities of classes. To this end, the authors develop a fractal-dimension-based measure of the spatial dispersion of samples for a class and adjust the prediction probabilities accordingly. They report significant improvements on several well-known benchmarks.

**Strengths:**

+ Calibration is a viable method for reducing the impact of class/data imbalance.
+ The gap identified by the authors (i.e., taking location distribution into account during calibration) is definitely promising.
+ The proposed calibration method taking into account prior distribution of samples across space as well as the fractal dimension is novel.

**Weaknesses:**

1. Although the proposed approach is novel, some of the individual steps are not sufficiently motivated. For example:

1.1. "Eq. 9 downweighs the classes that appear most uniformly and it upweighs the classes that appear less uniformly. In practice, this scheme enforces a centre bias for frequent classes and no bias for rare classes, as shown in Fig. 2-bottom-right. Intuitively, removing the bias from the rare classes is better than rectifying it because it produces balanced detectors and aligns better with the target distribution as shown in our ablation and our qualitative results." => It is not clear why this is enforced. Can we not have frequent classes that have some spatial bias (other than the center?) or that are uniformly sampled?

1.2. Figure 4d-e: The implication of weak correlation between n(y) and Phi(y) should not be surprising since the fractal definition in Eq 8 does not depend on the number of occurrences in a cell (since an indicator function is used). However, it is not clear why this is preferred.

2. Fractal dimension measures the uniformity of spatial distributions. Is this the best choice for this purpose, considering its limitations (not being able to take the limit on G, thresholding it and fitting a line etc.)? One could use many different methods available for measuring dispersion.

3. The extension to the binary case in Eq 11 appears ad hoc and is not justified properly.

**Questions:**

Please see Weaknesses.

---

### Official Review · Reviewer_BfmT · 2024-10-28

**Soundness:** 3
**Presentation:** 3
**Contribution:** 3
**Rating:** 6
**Confidence:** 3

**Summary:**

This paper proposed a new post-calibration method, named Fractal calibration (FRACAL), to solve the problem of data imbalance which widely exists in long tail object detection. In particular, this method considered the distribution of categories in image space by using fractal dimension.

**Strengths:**

1) This paper attempts to introduce the class-location dependency to post-calibration, which enhances the performance of rare categories.

2) This work fuses the space information by integrating the fractal dimension and optimizes the detection performance.

3) This is a plug-in-plug-out module, so it does not need extra training.

**Weaknesses:**

1) I feel confused about the description of Fig. 1. Why class prior ps(y) is used to align ps(y, u|x) with pt(y|x) rather than align ps(y|x) with pt(y|x)? Do you mean that space information has been considered by previous work just for training data?

2) Although the fractal dimension is introduced, the theoretical explanation of how the fractal dimension specifically affects the detection of rare categories can be more detailed. It is possible to further discuss how the fractal dimension improves the performance of category-position dependencies or provide some more intuitive explanations.

3) Although FRACAL is applied at the inference stage, calculating the fractal dimension may affect the inference speed. It is recommended to add relevant complexity analysis to evaluate the impact of fractal dimension calculation on inference efficiency and the performance of this method on resource-constrained devices.

4) More visualization examples of experimental results can be added to demonstrate the performance of FRACAL in rare category detection, such as the detection location distribution or confidence changes of certain categories, to enhance the intuitiveness of the method's effect.

5) Fig. 2 needs to be explained in more detail, especially for the bar chart located in the right part.

6) Why do different datasets and ablation experiments have different evaluation indicators? I suspect that other indicators are not listed because they are not as good as existing methods. If not, please explain the reason.

7) More minor issues should be more attention. I just list part of them:
(1) Figure 1-left should be Fig. 1-left.
(2) Eq 3 lost dot (Eq. 3).
(3) The sub-tables in Table 4 should appear in the order of the ablation experiments.
(4) In Table 4-e, please clearly show the G values ​​used in the final model.

**Questions:**

NA

---

### Official Review · Reviewer_ip87 · 2024-11-01

**Soundness:** 3
**Presentation:** 2
**Contribution:** 3
**Rating:** 6
**Confidence:** 4

**Summary:**

The paper proposes a novel approach, Fractal Calibration (FRACAL), for long-tailed object detection, addressing the common issue of imbalanced datasets where rare classes are often underrepresented. The authors introduce a post-processing method that adjusts logits using fractal dimensions, accounting for both class frequency and spatial distribution. FRACAL does not require training, can be integrated with various detection models, and shows strong performance improvements on rare classes across several benchmarks, including LVIS, COCO, and OpenImages.

**Strengths:**

1. This paper addresses an underexplored aspect in long-tailed object detection by introducing spatial information into post-calibration using fractal dimensions. This is innovative and may inspire new directions in handling data imbalance in object detection.

2. The proposed FRACAL method achieves significant improvements in performance for rare classes without retraining or additional inference costs, making it a practical tool for applications needing efficient solutions.

3. The authors conduct extensive experiments on multiple datasets (LVIS, COCO, V3Det, OpenImages), comparing FRACAL to state-of-the-art methods. The consistent performance gains demonstrate FRACAL's robustness and generalizability across different detection settings.

**Weaknesses:**

1. While the fractal dimension is a core component of the proposed method, the paper could benefit from a more detailed analysis on how different fractal dimension values influence performance across categories. This would add insight into the method's adaptability to datasets with varying spatial distributions.

2. The paper briefly discusses the limitations of grid-based approaches but does not thoroughly compare FRACAL against alternative spatial calibration methods. Including this would help contextualize FRACAL's advantages and trade-offs relative to other spatial calibration techniques.

3. The choice of parameters, such as the fractal dimension coefficient (𝜆) and log base (𝛽), might make FRACAL difficult to tune for practitioners who may not have expertise in spatial calibration. More guidance on parameter selection based on dataset characteristics could improve usability.

**Questions:**

See Weaknesses*

---

### Official Review · Reviewer_imsd · 2024-11-03

**Soundness:** 2
**Presentation:** 2
**Contribution:** 2
**Rating:** 3
**Confidence:** 5

**Summary:**

This paper proposes a post-training logit adjustment method designed specifically for long-tailed object detection. Its novelty lies in incorporating spatial location distributions directly into logit adjustment.

A brief background on logit adjustment: it is primarily used when you have an imbalanced / long-tailed training set but you care about class-balanced accuracy for inference, that is, you have a balanced test set. Suppose you have a trained model f, and you want to predict the category of input x. Normally, you do $\arg\max_y f_y(x)$, where $f_y(x)$ represents the logit for class y for input x. Logit adjustment modifies this by computing $\arg\max_y f_y(x) + \log p_{train}(y) - \log p_{test}(y)$. In short, (log) class priors are factored in at the logit level during inference.

This paper is proposing to integrate object location distribution into logit adjustment. So, it is basically replacing the $\log p$'s above with other distributions that are aware of the spatial distribution of object instances within an image. They achieve this through a multiscale box-counting method where boxes span a coarse-to-fine grid (e.g. from 2x2, 3x3 to NxN).

The authors report improved performance  on LVISv1, COCO, V3Det and OpenImages.

**Strengths:**

The paper tackles a relevant and important problem. Post-training calibration of object detectors is of practical importance. Integrating spatial location awareness into logit adjustment is a novel and interesting idea.

The authors report improved results on many datasets.

**Weaknesses:**

There are three major weaknesses:
1. Clarity: The paper is not well-written. Important pieces of information are not clear.
2. Substantiation of the main claim: I am not sure whether the main claim of the paper, which is incorporating spatial location distributions, is substantiated.
3. Performance comparisons: Experimental comparisons do not provide a clear indication whether we should use this technique with modern object detectors.

I explain them in detail below.

## Clarity:

- Equation 2, which actually presents the mathematical model for class-location dependence, is not explained well. There are certainly some independence assumptions among category, location and generic objectness here. Such a key expression should be explained in sufficient detail.
- L213: “generic object occurrence” -> what does this mean exactly?
- The rule to deal with rare classes (lines 321-330) is not clear to me at all. It is not clear how orange and blue dots are identified.
- Figure 1, at the point where it appears, is not understandable at all. It heavily relies on notation, which is not introduced yet. The right panel of Figure 1 is simply a fancy figure. It doesn’t convey useful/key information to the reader.
- Figure 2 is also not understandable. If the p(hat,u) on the left reflects ground-truth, how come the p(hat,u) on the right is a good prediction? Same with tiara.
- Figure 5 is not understandable. Reading the explanation in the caption and the paragraph in lines 522-528 several times, I still do not know what this figure proves.
- In general, the math notation is lousy. E.g. z_y is called a logit (L251) and z_y,u is called a “predicted proposal” (L296). It would help if the notation is made more rigorous. “Generic object occurrence” is ambiguous. Instead, defining concepts using object proposals (boxes, anchors, etc.) would be more useful.
- Usage of all \cite commands are wrong. Authors should look up how to use \citep and \citet.
- What is “pp”? Percentage points? Should have been explained in text.

## Substantiation of the main claim:

- The main claim of the paper is incorporating location information to the prior. But is there a discrepancy between train and test sets regarding location distributions? In L257, authors themselves write that object location distribution is the same in train and test sets.
- Does the box counting method really reflect the spatial location distribution of instances? I am not sure. It seems like a complicated way of counting instances. For example, it is hinted in Figure 2 that “hat” and “tiara” are objects that occur at the top portion of images. How is this location information taken into account by the box-counting method? A convincing ablation would be as follows: let’s create two classes of objects. For the first one, let all its instances be in the top part of the images, and for the second one, let all its instances be in the bottom part. Would FRACAL perform better than baseline or other detectors?
- Claims made about performance may not hold true if more modern object detectors are used (see below).

## Performance comparisons:

- SOTA comparisons are using old backbones (Resnet50, 101), therefore the performance is limited. On LVIS, the performance is around 50 AP (https://paperswithcode.com/sota/object-detection-on-lvis-v1-0-val). Methods from 2022 report 53.4 AP_b, 48.1 AP_m (Li, Mao, Girshick, He; ECCV2022). Since FRACAL is training free, it should be easy to apply it to more recent detectors and report results. This would give us a better picture on how effective the proposed method is.
- More detailed results on COCO would help, such as AP_small. Also, it should be easy to create AP_rare, AP_freq on COCO.

**Questions:**

- Can you report AP_s, AP_m, AP_l, AP_r, AP_c, AP_f on COCO?
- Can you report results using SOTA object detectors?
- In all the papers I know, AP_box is larger than AP_mask. In your results, they are very similar to each other, and even AP_box<AP_mask in one case (Table 2). How do you explain this?
- How do you explain your assumptions in Eq2?
- Does FRACAL capture the location distribution differences when the number of instances between two classes are the same? (my second point above in the “Substantiation of the main claim”).

---

### Note · Authors · 2024-11-14

I have read and agree with the venue's withdrawal policy on behalf of myself and my co-authors.